# TERRA: A Novel Biomarker of Embryo Quality and Art Outcome

**DOI:** 10.3390/genes12040475

**Published:** 2021-03-25

**Authors:** Maria Santa Rocca, Ludovica Dusi, Andrea Di Nisio, Erminia Alviggi, Benedetta Iussig, Sara Bertelle, Luca De Toni, Andrea Garolla, Carlo Foresta, Alberto Ferlin

**Affiliations:** 1Department of Medicine, University Hospital of Padova, 35128 Padova, Italy; mariasanta.rocca@aopd.veneto.it (M.S.R.); andrea.dinisio@unipd.it (A.D.N.).; luca.detoni@unipd.it (L.D.T.); andrea.garolla@unipd.it (A.G.); 2GENERA Veneto, GENERA Center for Reproductive Medicine, 36063 Marostica, Italy; ludodusi@hotmail.com (L.D.); iussig@generaveneto.it (B.I.); bertelle@generaveneto.it (S.B.); 3Clinica Ruesch, GENERA Center for Reproductive Medicine, 80122 Napoli, Italy; alviggi@generaroma.it; 4Department of Clinical and Experimental Sciences, Unit of Endocrinology and Metabolism, University of Brescia and ASST Spedali Civili Brescia, 25121 Brescia, Italy; alberto.ferlin@unibs.it

**Keywords:** telomeres, TERRA, lncRNA, sperm telomere length, ART

## Abstract

Telomeres are considered to be an internal biological clock, and their progressive shortening has been associated with the risk of age-related diseases and reproductive alterations. Over recent years, an increasing number of studies have focused on the association between telomere length and fertility, identifying sperm telomere length (STL) as a novel biomarker of male fertility. Although typically considered to be repeated DNA sequences, telomeres have recently been shown to also include a long non-coding RNA (lncRNA) known as TERRA (telomeric repeat-containing RNAs). Interestingly, males with idiopathic infertility show reduced testicular TERRA expression, suggesting a link between TERRA and male fertility. The aim of this study was to investigate the role of seminal TERRA expression in embryo quality. To this end, STL and TERRA expression were quantified by Real Time qPCR in the semen of 35 men who underwent assisted reproductive technologies (ART) and 30 fertile men. We found that TERRA expression in semen and STL was reduced in patients that underwent ART (both *p* < 0.001). Interestingly, TERRA and STL expressions were positively correlated (*p* = 0.010), and TERRA expression was positively associated with embryo quality (*p* < 0.001). These preliminary findings suggest a role for TERRA in the maintenance of sperm telomere integrity during gametogenesis, and for the first time, TERRA expression was found as a predictive factor for embryo quality in the setting of assisted reproduction.

## 1. Introduction

Infertility is a multifactorial disorder affecting one out of six couples in Western countries, and male factor infertility is implicated in about 50% of cases [1,2,3].

Although over recent years, well-defined risk factors associated with reduced male fertility have been found [4,5], the identification of novel biomarkers of infertility still remains a significant issue for important progress in global public health. Indeed, despite advances in the diagnosis of male infertility thanks to the advent of new tests, about one-third of the causes of infertility remain unexplained [6,7].

To date, the conventional procedures for the evaluation of male factor infertility include physical examination, semen analysis and endocrine and imaging studies. Furthermore, since spermatogenesis is a highly specialized process, the identification of genetic factors that are not detected by traditional tests and impact upon fertility assumes a particular importance [3].

Telomeres, which are repeated sequences of TTAGG at the end of mammalian chromosomes, are considered to be the internal biological clocks of cells. In fact, telomeres shorten with each cell cycle until they reach a critical length beyond which the cell undergoes senescence or apoptosis [8]. Telomerase is the enzyme deputed to restore the loss of telomeric DNA which occurs as a consequence of incomplete replication of the end of chromosomes. Telomerase activity is progressively turned off in somatic cells, except for within neoplastic cells and mature gametes [9,10,11]. Since telomeres are a genetic marker of biological age, it is evident that an abnormal and accelerated erosion leads to age-related diseases (O’Sallivan and [12]. Therefore, their preservation is a fundamental requisite for genome stability.

Recently, telomeres have been correlated with male fertility [13]; males with anomalies in semen parameters showed shorter sperm telomeres than normozoospermic males, resulting in a reduced fertilization rate and poor embryo quality [14,15,16]. Moreover, the important role of sperm telomere length (STL) in reproduction has also been confirmed by animal studies that have observed long telomeres exclusively in offspring conceived from male mice with long telomeres [17,18]. However, whether short sperm telomeres are the cause or the effect of male infertility is a topic that still remains to be elucidated [16].

For several years, it has been assumed that telomeres consisted exclusively of repeated DNA sequences and shelterin proteins, a complex of proteins implicated in the recruiting of telomerase and in the protection of the end of chromosomes from DNA repair machinery. However, recent studies have highlighted a long non-coding RNA (lncRNA) localizing on the chromosome caps, referred to as TERRA (Telomeric Repeat-containing RNA) [19,20]. Reig-Viader et al. have investigated the role of this lncRNA in spermatogenesis, observing a higher expression of the transcript during the crucial phase of meiosis in which crossing-over and chromosomal segregation occurs [21]. The findings of this study that show the importance of TERRA during spermatogenesis have been further confirmed with the detection of more frequent testicular TERRA foci in a fertile male that underwent vasectomy, compared with the testes of four azoospermic men [22].

Based on the hypothesis of a likely contribution of TERRA in male fertility, we aimed to evaluate TERRA expression in the semen of men undergoing ART (assisted reproductive technology) in order to investigate whether a link between this lncRNA and embryo quality exists with consequential successful outcomes in ART. 

## 2. Materials and Methods

### 2.1. Participants

We prospectively included 35 men from couples who underwent their first fresh intra-cytoplasmic sperm injection (ICSI) treatment and 30 volunteers with proven fertility as controls (male partners of couples with successful pregnancy within the first 12 months of regular unprotected sexual intercourse). The exclusion criteria for cases were evident causes of spermatogenic impairment such as seminal infection, sperm autoantibodies, varicocele, history of cryptorchidism, orchitis, testicular cancer, anticancer treatments, karyotype anomalies or Y chromosome long arm microdeletions. The exclusion criteria for the female patients were a history of ovarian surgery or medication known to impact ovarian function.

Of the 35 males included in the study, 4/35 (11%) had oligoasthenoteratozoospermia (OAT) and 31/35 (89%) had normozoospermia. Of the women, 24 had female factors such as advancing maternal age, tubaric factor or reduced ovarian reserves, and 7 were couples with idiopathic infertility.

This study was approved by the University Hospital of Padova. All participants signed the informed consent form.

### 2.2. Assessment of Oocyte Survival, Fertilization and Further Development

Normally fertilized oocytes (zygotes) were cultured in G1.2 medium (Vitrolife, Gothenburg, Sweden) at 37 °C and equilibrated with 5% CO_2_ in air. Fertilization was assessed 16–18 h after ICSI according to previously published criteria [23]. Cleaving embryos were evaluated on day 3 after ICSI with the use of a cumulative embryo classification scheme taking into account cleavage speed, blastomere symmetry, extent of fragmentation and the presence or absence of multinucleated blastomeres [24].

### 2.3. Semen Analyses

Standard semen analysis was performed according to the World Health Organization protocol (WHO 2010) after 2–7 days of sexual abstinence. Briefly, semen volume was measured by weight, assuming a semen density of 1.0 g/mL; sperm concentration was evaluated by a hemocytometer (Burker–Turk; Paul Marienfeld GmbH & Co. KG, Lauda-Königshofen, Germany); sperm morphology was identified from semen smears prepared with 10 mL of well-mixed semen, stained with Papanicolaou and assessed using the Tygerberg strict criteria. Sperm motility was graded into total (progressive + non progressive motility) and progressive motility. Total sperm count (volume x sperm concentration) was also calculated. Sperm cells were isolated by Percoll (Sigma-Aldrich, Darmstadt, Germany) 45% and 90% gradients. For this study, the remaining sperm after insemination was divided in two aliquots in order to measure STL and TERRA expression.

### 2.4. Measurement of STL in Semen

To measure STL, genomic DNA was isolated from the sperm of the ART group and the healthy controls using a QIAamp DNA Blood Mini Kit (Qiagen, Hilden, Germany) according to the manufacturer’s instructions. Telomere length was measured using a quantitative real-time PCR method according to the technique established previously [25], and 36B4 was used as a control. The relative telomere length was calculated as Telomere/36B4 ratio using 2^−△Ct^ (△Ct = Ct telomere−Ct 36B4). The primers used for qPCR have been previously described [25]. Reactions were run in the Step One Real-Time PCR System (Applied Biosystems, California, USA). All the samples were processed in triplicate.

### 2.5. Quantification of TERRA Expression

Genomic RNA was isolated from the sperm of the ART group and healthy controls by Trizol (ThermoFisher Scientific, Massachusetts, USA), treated with DNasel and digested again with DNase on the column (Qiagen RNase-Free DNase Set; Qiagen, Hilden, Germany). RNA purity and concentration was measured by a NanoDrop^®^ spectrophotometer (ThermoFisher Scientific, Massachusetts, USA) and Qubit Fluorometric Quantitation (ThermoFisher Scientific, Massachusetts, USA). Total TERRA transcript RNA samples were converted to cDNA by SuperScript III Reverse Transcriptase (ThermoFisher Scientific, Massachusetts, USA) according to the manufacturer’s protocol using 2 pmol of specific oligonucleotides published previously (TERRA-specific RT primer: CCCTAACCCTAACCCTAACCCTAACCCTAA; GAPDH-specific RT primer: GCCCAATACGACCAAATCC) (Feretzaki and Lingner 2017). After cDNA synthesis, a real-time PCR on Step OnePlus Real-Time PCR System with SYBR^®^ Select Master Mix was performed (Applied Biosystems) using published primers that showed a high amplification efficiency (9p forward: GAGATTCTCCCAAGGCAAGG–9p reverse: ACATGAGGAATGTGGGTGTTAT; XpYp forward: GCGCGTCCGGAGTTTG–XpYp reverse: CCACAACCCCACCAGAAAGA) [26,27]. The GAPDH gene was used as the reference control (GAPDH forward: AGCCACATCGCTCAGACAC-GAPDH reverse: GCCCAATACGACCAAATCC). For each qPCR sample, three technical replicates were carried out with the following conditions: 95 °C for 15 s, 60 °C for 1 min for 40 cycles, followed by the dissociation stage for melting curve analysis. Relative quantities of TERRA were calculated as the ratio of TERRA to GAPDH using 2^−△Ct^ (△Ct = Ct TERRA − Ct GAPDH).

### 2.6. Statistical Analysis

Statistical analysis was performed with SPSS software, version 21.0 (SPSS Inc., Armonk, NY, USA). Data were reported as means ± standard deviation (SD) of the mean. The Kolmogorov–Smirnov normality test was used to detect the normality of the data. The Student *t*-test was used to compare data between cases and controls. The association between TERRA and semen and embryo parameters was analyzed with Pearson correlation analysis or Spearman correlation analysis for non-normally distributed variables. The significance level was set at *p* value < 0.05.

Based on the correlation analyses, we then performed stepwise multivariate regression analysis to evaluate the impact of independent variables on embryo quality. The significance level for entering and for removal of variables from the model was *p* = 0.05 and *p* = 0.10.

## 3. Results

Blastocyst transfer was performed on only 7 out of the 35 couples included in the study, whilst it was cancelled in those remaining due to genetic anomalies detected by preimplantation genetic diagnosis (PGD). Of the seven blastocyst transfers, only four, falling within the couples whose male partner had normal semen parameters, resulted in clinical pregnancies and successful deliveries.

Real-time qPCR data were obtained exclusively using 9p primers, since data from the mix including XpYp primers were excluded because of the formation of primer dimers. 

Table 1 shows the main characteristics in seminal parameters, telomere length and TERRA expression between the ART group and controls. TERRA expression and STL were significantly lower in the ART group compared with controls (*p* < 0.001). Although the two groups had a statistical significance in age, age did not correlate with TERRA expression, thus excluding an effect of age on TERRA levels.

A positive and statistically significant correlation between TERRA expression and embryo quality (r = 0.655, *p* < 0.001) (Figure 1) and STL (r = 0.428, *p* = 0.010) was found (Figure 2).

Based on correlation analyses, we then performed separate multiple linear stepwise regression analyses using the variables that proved significant in bivariate correlations as possible independent variables in predicting embryo quality. TERRA expression showed to be the only independent predictor of embryo quality in the ART group (β = 0.578; t = 4.068; *p* < 0.001), accounting for 33.1% of the variance in embryo quality.

## 4. Discussion and Conclusions

Telomeres are structures with a ribonucleoprotein capping at the end of eukaryotic chromosomes and have the function to protect chromosomes from degradation, erroneous recombination and fusion. They consist of tandem repeat DNA sequences (TTAGGG)_n_, TERRA and shelterin complex. The latter is composed of six proteins that prevent erroneous DNA damage responses and recruit telomerase enzymes at the 3′ ends of chromosomes [28,29,30].

Due to end-replication problems of chromosome ends, the telomeres shorten at each cell division until they reach a critical length leading to cell senescence. Furthermore, telomere length can also be influenced by other causes such as altered telomerase activity, genetic defects and stress agents, all factors contributing to telomere erosion and, consequently, to genome instability [31].

Human telomeres generally range in size from 8 to 15 kb [32] with the exception of germ and neoplastic cells presenting longer telomeres than somatic cells. Since telomerase activity gradually decreases in most somatic cells, an alternative lengthening of telomeres (ALT) through a telomerase-independent manner based on a homologous recombination (HR)-mediated process could explain the different telomere length between somatic and germ line cells [33,34].

Over the last decade, many studies have shown the relationship between telomere length and fertility, suggesting that dysfunctional telomeres could negatively affect human reproduction, resulting in reduced fertilization rate and poor embryo quality [15]. However, it is still not clear whether telomere shortening is the cause or the effect of decreased fertility [35].

Telomeres consist of constitutive heterochromatin; hence, it has been assumed for many years that they were transcriptionally silent. However, it has recently been found that a lncRNA, known as TERRA, is transcribed from subtelomeres towards chromosome ends [19].

TERRA ranges in size from 100 bases to 100 kb, and a growing body of evidence reports that it is implicated in telomere homeostasis [19,20,36]. Although the exact mechanism of action of TERRA still remains to be elucidated, its role in preventing telomere dysfunction seems be a fundamental point [37,38]. Since functional telomeres ensure the correct separation of chromosomes during meiosis [39], the maintenance of telomere length by TERRA could be essential in order to generate gametes with normal chromosomal structure.

Considering that TERRA is a fundamental component of telomeres, the hypothesis of its implication in human reproduction is very likely. Based on this plausible association between TERRA and fertility, the expression and localization of this lncRNA have been investigated in subjects with reduced fertility [22,40]. Based on recent evidence reporting low TERRA levels in the testes of infertile subjects, we aimed to evaluate the presence of this lncRNA in mature sperm and its expression levels in the sperm of 35 males from couples who underwent ART.

We found lower TERRA expression in the sperm of the ART males than in controls. Intriguingly, there was a positive and statistically significant association between TERRA and STL, and more importantly, between TERRA expression and embryo quality, suggesting that this lncRNA in sperm could be an important predictor of embryo quality in the setting of ART. The latter finding is very interesting because it could be the result of reduced TERRA on sperm DNA quality, translating into an impaired embryonic development. Although we have no data about sperm DNA fragmentation, the hypothesis of increased frequency of DNA damage in males with reduced TERRA could be likely. We have previously reported a positive and statistically significant correlation between STL and sperm quality parameters such as DNA fragmentation and chromatin status in normozoopsermic males [16]. We could, therefore, speculate that an association of TERRA levels with non-conventional sperm parameters might be plausible. This hypothesis is corroborated by the findings of our group [16] that show lower DNA fragmentation in individuals with long STL and normal sperm motility, suggesting that functional telomeres have lower DNA defects. Indeed, we observed that longer STL corresponded to greater expression of TERRA in sperm, further supporting a relationship between these two parameters with important effects not only on DNA integrity and semen quality, but also with successful fertilization and subsequent embryo development. Therefore, based on our results, we could suppose that subjects showing low TERRA expression could have a higher risk of segregation errors during meiosis resulting in dysfunctional sperm telomeres that could impact on egg fertilization and embryo quality.

Since the intra-cytoplasmic sperm injection (ICSI) procedure selects the spermatozoa only on the basis of morphology and motility parameters, the possibility of introducing sperm with genome instability into the oocyte is very high, thus increasing the risk of fertilization failure. Therefore, since the oocyte is incapable of repairing high DNA damages carried from fertilizing spermatozoon [41,42], high-quality male gametes are indeed essential to generate high-quality embryos [43,44].

In conclusion, this is the first study evaluating TERRA expression in mature human spermatozoa and pointing out lower TERRA expression in male partners of couples undergoing ART compared to fertile males. The finding of reduced TERRA levels in these subjects could suggest an important contribution of TERRA to fertilization potential. Although our results allow no conclusion on the overall impact of TERRA on reproductive success, they identify this transcript as a predictive factor of sperm and embryo quality. The major limitations of the present study include the small sample size of the subjects and the lack of information about their sperm DNA integrity. Furthermore, since good embryo quality also depends on oocyte quality, we cannot establish with absolute accuracy whether the low good embryo quality found in the couples with female factor as diagnosis and male partners with low TERRA is also due to oocytes. Therefore, it is essential for future studies to exclusively select individuals with male factor or idiopathic infertility in order to better address the function of TERRA in successful fertilization outcomes.

However, although the function of this lncRNA needs to be confirmed in further studies to explore these findings in further detail, this is the first study reporting the relationship between seminal TERRA expression and embryo quality from ART. This interesting result could suggest the evaluation of TERRA as a novel seminal biomarker in fertility treatments in order to increase the probability of successful outcomes in ART.

## Figures and Tables

**Figure 1 genes-12-00475-f001:**
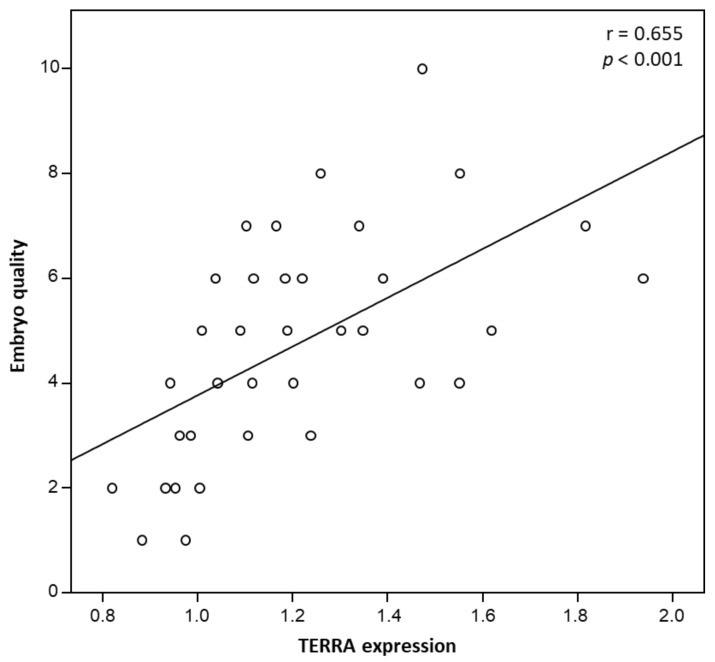
Correlation between 9p sperm TERRA expression and embryo quality after ART.

**Figure 2 genes-12-00475-f002:**
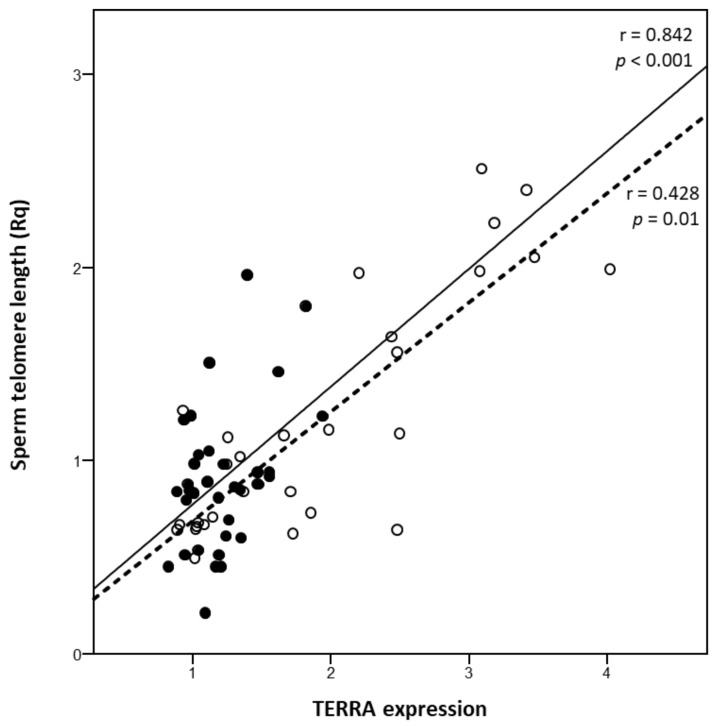
Correlation between sperm TERRA expression and sperm telomere length (STL) in ART patients (full circles) and controls (empty circles, dashed line). STL is expressed as Telomere/Single-copy gene ratio (T/S).

**Table 1 genes-12-00475-t001:** Relative telomere length, TERRA and clinical parameters in males who underwent ART and control group.

Parameters	ART group (*n* = 35)	Controls (*n* = 30)	*p*
Age (years)	39.± 6 5.4	36.1 ± 6.8	**0.02**
Sperm concentration (cells/mL ×10^6^)	56.8 ± 35.3	59.2 ± 57.5	0.84
Total sperm count (×10^6^)	161.2 ± 110.9	159.9 ± 127.35	0.96
Motility (%)	44.5 ± 13.8	43.5 ± 13.5	0.77
Vitality (%)	80.5 ± 10.1	81 ± 7	0.83
Morphology (%)	6.1 ± 4.3	7.6 ± 5.1	0.21
Sperm telomere length (T/S)	0.9 ± 0.3	1.2 ± 0.6	**0.02**
TERRA expression	1.2 ± 0.2	1.8 ± 0.9	**<0.001**

Significant *p* values are in bold. Data are expressed as means ± SD. ART: assisted reproductive technology.

## Data Availability

The data presented in this study are available on request from the corresponding author. The data are not publicly available due to privacy.

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
