# Peer review of "TERRA: A Novel Biomarker of Embryo Quality and Art Outcome"

_genes, 2021, doi:10.3390/genes12040475_

Round 1
Reviewer 1 Report
In this manuscript, Rocca et. al. demonstrated a positive correlation between TERRA expression in semen and embryo quality, suggesting that TERRA may be a biomarker for embryo quality. This finding is interesting and may help improve the success rate of assisted reproduction technologies. There are some critical points listed below that are needed to be addressed before publication.
- The authors claim that there is a significant association between TERRA expression and STL (sperm telomere length) in the abstract. But the authors only show that both levels were reduced in the ART group. The author could show the correlation plot similar to Fig. 1 for TERRA expression and STL.
- Is the result of TERRA expression by qRT-PCR only for TERRA transcripts derived from chromosome 9 (as the authors mentioned XpYp primers were excluded)? If so, please specify that in the main text and Fig. 1. TERRA derived from different chromosome ends could be regulated differently according to cell types and various stimuli.
- On page 4, the authors mentioned that telomere contains TTAGGG repeats at the end of the eukaryotic chromosomes. This sentence should be correlated because yeast cells contain different telomeric repeat sequences. Mammalian telomeres consist of TTAGGG repeats.
- On page 5 (line 199), the authors mentioned that the size of human telomeres generally ranges from 5 to 15 kb. The normal human telomere length should be around 8-15 kb according to the paper they cited.
- On page 6, the references the authors cited for this statement “the role of TERRA in preventing telomere dysfunction seem to be a fundamental point” are Wang et al. 2015 and Koch 2017. It is better to cite the original research article published in Cell by Chu et al in 2017. TERRA RNA Antagonizes ATRX and Protects Telomeres.
Cell 2017 Jun 29;170(1):86-101
The paper published by Koch in 2017 is a review article for new findings.
- On page 6 (line 263). Furthermore, we are unable to evaluate in those couples with a diagnosis of female factor infertility the contribution of oocyte to embryo quality.
This sentence is unclear. Please rephrase it.
Grammatical errors: I have suggested the correlation in parentheses.
- Abstract (line 24): TERRA and STL expression was (were) positively correlated.
- Figure 1. Correlati on (Correlation) between s perm (sperm) TERRA expression
- Line 32. Infertility is a multifactorial disorder affecting one out of six couples in Western countries and male factor is implicated (in) about 50% of cases of infertility.
- Line 36. the identification of 35 novel biomarkers of infertility still remains a significant issue for important progresses(progress) in global public health.
- Line 45. Telomeres, that (which) are repeated sequences of TTAGG at the end of the eukaryotic (mammalian) chromosomes.
- Line 69. However, recent studies have highlighted a long non-coding RNA (lncRNA) localizing on the chromosomes (chromosome) caps, referred to as TERRA.
- Line 164. Of the seven blastocyst transfer (transfers), only four, 164 falling into the couples whose male partner (partners) had normal semen parameters,
- Line 172. Although the two groups had a statistically significant different age (a statistical significance in age), age did not correlate with TERRA expression, excluding thus an effect of age on TERRA levels.
- Line 213. however, recently it has been found that a lncRNA, known as TERRA, is transcribed from subtelomeres towards chromosomes (chromosome) ends
- Line 217. Although the exact mechanism of action of TERRA still remains to be elucidated, its role in preventing telomere dysfunction seem (seems to) be a fundamental point.
- Line 224. the expression and localization of this lncRNA has (have) been investigated in subjects with reduced fertility
Author Response
- The authors claim that there is a significant association between TERRA expression and STL (sperm telomere length) in the abstract. But the authors only show that both levels were reduced in the ART group. The author could show the correlation plot similar to Fig. 1 for TERRA expression and STL.
Answer: We added Figure 2 showing the correlation between TERRA expression and STL as Reviewer suggested.
2. Is the result of TERRA expression by qRT-PCR only for TERRA transcripts derived from chromosome 9 (as the authors mentioned XpYp primers were excluded)? If so, please specify that in the main text and Fig. 1. TERRA derived from different chromosome ends could be regulated differently according to cell types and various stimuli.
Answer: The Reviewer is correct, we obtained qRT-PCR data using primers targeting chromosome 9p. We added it in Result and Fig. 1.
3. On page 4, the authors mentioned that telomere contains TTAGGG repeats at the end of the eukaryotic chromosomes. This sentence should be correlated because yeast cells contain different telomeric repeat sequences. Mammalian telomeres consist of TTAGGG repeats.
Answer: The Reviewer is correct, we rephrased the sentence as she/he suggested.
4. On page 5 (line 199), the authors mentioned that the size of human telomeres generally ranges from 5 to 15 kb. The normal human telomere length should be around 8-15 kb according to the paper they cited.
Answer: We apologize for typo. We corrected telomere length as reported from bibliographic reference.
5. On page 6, the references the authors cited for this statement “the role of TERRA in preventing telomere dysfunction seem to be a fundamental point” are Wang et al. 2015 and Koch 2017. It is better to cite the original research article published in Cell by Chu et al in 2017. TERRA RNA Antagonizes ATRX and Protects Telomeres.
Cell 2017 Jun 29;170(1):86-101
The paper published by Koch in 2017 is a review article for new findings.
Answer: We added Chu et al. in the text and bibliography as Reviewer suggested.
6. On page 6 (line 263). Furthermore, we are unable to evaluate in those couples with a diagnosis of female factor infertility the contribution of oocyte to embryo quality.
This sentence is unclear. Please rephrase it.
Answer: We clarified the sentence as Reviewer suggested.
Grammatical errors: I have suggested the correlation in parentheses.
- Abstract (line 24): TERRA and STL expression was (were) positively correlated.
- Figure 1. Correlati on (Correlation) between s perm (sperm) TERRA expression
- Line 32. Infertility is a multifactorial disorder affecting one out of six couples in Western countries and male factor is implicated (in) about 50% of cases of infertility.
- Line 36. the identification of 35 novel biomarkers of infertility still remains a significant issue for important progresses(progress) in global public health.
- Line 45. Telomeres, that (which) are repeated sequences of TTAGG at the end of the eukaryotic (mammalian) chromosomes.
- Line 69. However, recent studies have highlighted a long non-coding RNA (lncRNA) localizing on the chromosomes (chromosome) caps, referred to as TERRA.
- Line 164. Of the seven blastocyst transfer (transfers), only four, 164 falling into the couples whose male partner (partners) had normal semen parameters,
- Line 172. Although the two groups had a statistically significant different age (a statistical significance in age), age did not correlate with TERRA expression, excluding thus an effect of age on TERRA levels.
- Line 213. however, recently it has been found that a lncRNA, known as TERRA, is transcribed from subtelomeres towards chromosomes (chromosome) ends
- Line 217. Although the exact mechanism of action of TERRA still remains to be elucidated, its role in preventing telomere dysfunction seem (seems to) be a fundamental point.
- Line 224. the expression and localization of this lncRNA has (have) been investigated in subjects with reduced fertility
Answer: We corrected grammatical errors according to Reviewer suggestions.

Reviewer 2 Report
This work is the first study evaluating TERRA expression in mature human spermatozoa and pointing out lower TERRA expression in male partners of couples undergoing ART compared to fertile males. The finding of reduced TERRA levels in these subjects could suggest an important contribution of TERRA in fertilization potential. Although results allow no conclusion on the overall impact of TERRA on reproductive success, they identify this transcript as a predictive factor of sperm quality but not embrio quality. The major limitations of the present study include the small sample size of the subjects and the lack of information about their sperm DNA integrity.
Author Response
We thank the Reviewer for her/his comment. As Reviewer wrote, our results allow no conclusion on the overall impact of TERRA on reproductive success and need further studies as we claimed. However, it is the first study evaluating TERRA expression in mature human spermatozoa and measuring TERRA in male partners of ART couples.
This manuscript is a resubmission of an earlier submission. The following is a list of the peer review reports and author responses from that submission.
Round 1
Reviewer 1 Report
The manuscript titled “TERRA: A Predictive Biomarker of Male Fertility and Embryo Quality” is an interesting study. Here the authors used the telomere length and long noncoding RNA – TERRA expression as biomarkers for male infertility and ART outcomes. Although the study question and the topic of research is current field of interests, I have concerns with the design of the study that is detailed below.
This study includes subjects who are undergoing ICSI treatment (n=35) and fertile controls (n=30). 31 of the infertile couple had normal semen analysis and 24 of them had female infertility issues. It is inappropriate to consider this patient group as with men with infertility issues.
The sample size is too small to statistically come to a meaningful conclusion.
Although sperm telomere length and TERRA expression were statistically significant with embryo quality on day 3, this was not validated by further embryo development as only 7 couples proceeded to blastocyst resulting in 4 pregnancies.
The discussion part is poorly written.
Reviewer 2 Report
Review for manuscript from Rocca et al untitled : “TERRA: A Predictive Biomarker of Male Fertility and Embryo 2 Quality”
Authors claim they identified a new biomarker to predict ART outcomes. Everyone is desperate to find such biomarkers. Unfortunately, the way this work has been conducted does not allow any conclusion.
The author are aware of this since in their conclusion they give the reasons to reject their paper, I quote them : “The major limitations of the present study include the small sample size of the subjects and the lack of information about their sperm DNA integrity. Furthermore, we are unable to evaluate in those couples with a diagnosis of female factor infertility the contribution of oocyte to embryo quality. Therefore, it is essential to select exclusively individuals with male factor or idiopathic infertility in order to better address the function of TERRA in successful fertilization outcomes.”
Indeed, the patients recruitment introduce to many confusing elements, female infertility factors including maternal age, which could affect the expression of TERRA, statistically significant age difference between the group studied (not mentioned in the text), which could explain the difference of expression of TERRA and the difference of embryo quality.
In addition, it is not clear if the patient and control samples have been treated the same way and how. Any tiny differences could explain a difference in the level of expression of TERRA. Authors claimed a correlation between sperm TERRA expression and embryo quality, but they don’t mention where are situated the blastocystes that give to pregnancy. It is worth noting that the global IVF results are low to very low.